# Divorce: Experiential and Structural Elements: Cases from Papua New Guinea and Africa

**Pamela J. Stewart \* and Andrew J. Strathern \***

Department of Anthropology, University of Pittsburgh, Pittsburgh, PA 15260, USA
\* Correspondence: ajstuden@pitt.edu (P.J.S.); strather@pitt.edu (A.J.S.)

**Abstract:** Divorce emerges as a phenomenon in counterpoint to marriage, both terms representing processes or phases of interaction punctuated by moments of completion and transition to further phases. We can make an initial distinction between divorce, viewed as undoing of preceding phases, and marriage, viewed as prospective of moving into a new relationship. Both divorce and marriage may carry different meanings depending on the wider culture in which they occur. Where marriage comes into being via a series of reciprocal transactions of wealth objects, divorce correspondingly consists of the undoing of such transactions, with the aim of creating a new order of relationships. This process can, in turn, itself vary as it turns on emotional manifestations between the parties involved, sometimes connected with the presence of offspring, as in the case of the Nuer people of South Sudan, among whom a wife does not shift to her husband's settlement place until the couple have a child. The question of transactions goes with the significance of the wider kin networks in which marriages and divorces are regulated. All in all, our paper examines a counterpoint between legal and emotional aspects of both marriage and divorce, raising issues about what a marriage is and what constitutes a divorce, together with nuances of ritual processes that mark pathways between these categories. We draw on ethnography from Pacific cultures, especially Papua New Guinea, and from Africa, to explore these processes.

**Keywords:** conjugal fund; Hagen; bridewealth; adultery; village courts; churches; Nuer; Nayar; cattle; polygyny; gender

## 1. Introduction

The term 'divorce' applies to the termination of a marital relationship by agreement or imposition. It involves the undoing of the kinds of ties that constitute the marital relationship itself, together with concomitant personal feelings and the objective legal factors that flow from the original marital ties and assignments for the divorce. Divorce is a strong term, entailing recognition of severance between the parties involved. Divorce procedures constitute a ritual of unmaking that reverses the processes of making or creating that constitute marriage. Just as marriage legally brings into being a whole set of circumstances that convert an experiential situation into a normative one, divorce also does this by cancelling the ties so constituted. Looking at the scenario from the point of view of ritual theory, we can invoke here Roy Rappaport's maxim about ritual: it turns analog relationships into digital ones (Rappaport 1968). Analog relationships in this context contain the affective experiential senses that help to build the base for their transformation into legal ties that emanate from religious or state sources. Of course, while the digital/analog distinction is clear-cut, in actual experience, the matter is not so simple because analog-derived features can persist into an altered digital structure and quasi-legal aspects can enter into an experiential context, such as when a couple decide to become 'engaged' to be married or when, in arranged marriage contexts, two people are affianced before they are of age to be married. Informal marriage, permitting sexual relations to commence, also happens, as in the old Scottish 'hand-fasting' ritual.

The general point here is that much variation can be discerned as elaboration on or deviance from a digital structure. 'Love' may be invoked, and 'love' can also be variable, but, in systems where 'love' is considered a crucial diagnostic of a working sexual relationship or marriage, a declaration that 'love' is no longer present between the partners marks the potential dissolution of the marriage in itself. Either partner or both may make such a declaration as a reason why the partners can now look elsewhere for a mate. 'Breakdown' of the marriage is implied.

## 2. Dowry and Bridewealth Societies

These remarks we have made are deliberately couched in categories that apply in a familiar way within Euro-American cultural parameters. The validity of these categories is partly ethnocentric and depends on underlying structures and values that come to define 'the family'. British anthropologist Jack Goody developed a theoretical scheme that helps to put this point in analytical terms (Goody 1983). He identified a pattern of inheritance that he called 'diverging devolution', in which family resources are parceled out between siblings of both sexes and each marriage ideally brings together two sets of resources, one from the female partner, one from the male. The endowment of female inheritors can take the form of dowry, and this dowry makes up, along with whatever the male partner inherits, a resource that Goody calls the conjugal fund. Joining of resources in this way tends to solidify the marital bond. Concomitantly, if and when a marriage is dissolved by divorce, negotiations between the parties involve disposition of the conjugal fund and assessments of the economic value of the assets of the pair, distinguishing assets brought into the marriage and those jointly created during it. Conflicts over these resources arise from the idea of a conjugal fund existing in the first place.

Goody makes a contrast between 'dowry societies' and 'bridewealth' societies. We use here the example of Melpa speakers in the Mount Hagen area in the Western Highlands of Papua New Guinea (Strathern and Stewart [2011] 2016, 2000a, 2000b; Andrew Strathern 1980). In Hagen, and in bridewealth contexts generally in the Western Highlands of Papua New Guinea, a husband gains control over aspects of the wife's person, with particular reference to reproductive capacities flowing from rights of sexual access. The wife's kin may make a counter-prestation by endowing her with some pigs to rear and care for, but this is not a form of inheritance, only a start for her new role as a producer of value for the husband's group. If divorce occurs later, this endowment does not form a part of remunerations among those involved. What does produce conflict is what to do about the bridewealth payments that customarily constitute the marriage and continue to do so even if the couple are estranged and the wife goes back to her own kin. Therefore, in this case, the rituals of divorce literally reverse the rituals of exchange that constitute the marriage.

Accordingly, the rituals of divorce are the negation of the exchanges that constituted the bridewealth. Making this negation definitive depends on meticulous accounting among the relations of the couple. A precise and agreed recall of all the items at stake is necessary. Participants must be prepared to sit down together and, in spite of hard feelings, thrash out each detail. Sometimes, one item is declared to have been repaid earlier with a similar item. Sometimes, another item may have been substituted, by agreement, between the parties. In bridewealth contexts, items may be reciprocated for each other at the time of the marriage itself, in which case these will not enter into divorce discussions Moreover, in these kinds of societies, marriage is simply one nodal point of exchange relations. People may build further exchanges along the pathway of the marriage, and these exchanges could theoretically continue in spite of a divorce, but, in practice, this is not likely to occur because the divorce itself brings ill-will and distrust. All exchange partnerships are supposed to depend on trust, and the basis for trust is broken when a divorce happens. In Mount Hagen, partners in exchange relationships are supposed to have min, good feelings, closeness, in the Melpa language, and divorce tends to break up such feelings. Blame may be laid on the father or mother of the wife/bride for trying to obtain too much bridewealth, and there is an expression in the Hagen language to indicate that a father may withdraw his

daughter from a marriage, with the assumption that a father's influence can bring this about (mboklam mõngi). This indicates that the interpersonal tie between spouses is seen as based on choice and can be changed in the face of pressure from kin.

In bridewealth societies, the dominant emphasis is on maintenance of exchanges and a reliance on the human effort needed to achieve this. Women as wives participate in this ideology, but only to the extent that they feel the exchanges are worthwhile for them, most notably if wealth flows to their own natal kin. If this is not, or is no longer, the case, there is the possibility for the marital tie to be weakened. The presence of children can ameliorate a situation such that the life-cycle rituals centered on the children can prove a channel for supporting the original marriage.

Polygyny can also affect a marital tie, and polygynous men face difficulties in satisfying the wishes of multiple partners. The wives also experience frustration and anger because of the competitive character of their overall situation. Many factors, then, influence the stability or instability of a marriage. The dominant ideology in these societies is based on an idea of equity, treating spouses fairly. If equity is violated, the partners will become dissatisfied. A model for equity here has to do with allotment of strips of garden land. A male householder marks strips of land for spouses and dependents to cultivate, planting and harvesting crops for subsistence or sale. As with all resources, this idea requires effort and planning, which are at the heart of all social action. Effort, in turn, is acceptable when it results in equity. The ideology of sharing fairly and distributing fairly underlies the cultural ideals of practice for both men and women, described as mok roromen. Failure to keep to this ideal between spouses can, over time, lead to divorce procedures.

The rituals defining divorce are less prominent than the rituals of marriage. Marriage announces a change in social structure with the creation of new channels of exchange. The rituals of divorce in which this structure is undone tend to be pragmatic and low-key, stopping the flows of exchange, laying the conditions for a new configuration and wiping out the effects of the initial marriage exchanges.

In societies with bridewealth, then, divorce means dismantling of exchanges surrounding a marital pair. In dowry or conjugal fund societies, divorce means dismantling of property-sharing relations created by a marriage. In both cases, the precursor or trigger of the process of dissolution is a putative breakdown in interpersonal relations between the spouses. Sexual relations of cohabitation are routinely cited in this connection in conjugal fund societies. This domain is often an official part of the declaration of a process of breakdown, although this is not necessarily a causative circumstance. Fault in divorce was in the past associated with adultery in conjugal fund societies, later replaced in some cases by the concept of 'no fault' divorce. Cases are also always complex and indeterminate in themselves. A crucial point is reached if either party seeks to set up a new relationship, in which sexual relations are again brought into causative prominence. Mutatis mutandis, something similar can happen in bridewealth societies, typically by a wife leaving her residence with her husband and either going back to her parental home or moving on to a new partner or partners in an experimental mode if she is free to do so. A divorced woman commands a smaller bridewealth if she remarries, so there is usually pressure for a woman to go back to the first husband, but it is also recognized that this may not work. Property relations are not at stake in any of these cases, whereas, in conjugal fund societies, they are always central. In experiential terms, 'breakdown' of relationship may develop in a similar manner across this structural dichotomy, but the consequences are structurally determined. In turn, dispositions of property themselves lend to emotional distress and arguments that can lead to physical fighting as well as verbal recriminations. In bridewealth societies, recriminations tend to center around the exchanges that set up the marriage, and networks of kin get involved.

Outside of such a context of bridewealth, a partner in a conjugal fund marriage may take hold of an item of property belonging to their counterpart and forcibly throw it out, thereby pre-empting any debate or demurral. In cases we know, such action is taken when the couple had been living uxorilocally and the estranged wife throws out belongings

of her husband in lieu of discussion. Invariably, this happens in connection with the wife's complaint or anger that the husband has entered a sexual relationship with another woman. The wife is then telling him by her actions that he cannot continue in a privileged relationship with her, including storage of property items. Shared property space is an index of harmony, or at least accepted cohabitation. If that condition of mutuality is no longer operative, the property of the partner or ex-partner falls into the category described by anthropologist Mary Douglas as 'dirt' or 'matter out of place' (Douglas 1966) on 'Purity and Danger'. As such, it can be 'cleaned out' and put into a place designed for rubbish. Once, we saw an example in Ayrshire in Scotland that appeared to fit this circumstance in a lay-by along a road that had a rubbish container. There, among old bits of clothing, was a set of golf clubs still in their bag, seemingly pitched out by an angry spouse or partner no longer willing that these sport artifacts should stay in their accustomed position of recognition or toleration.

Throwing out property represents a strategy of self-help rather than waiting for an authoritative ruling by some influential entity. Such rulings are a hallmark of state-based societies in which values are, or can be, calculated and calibrated. Procedures of giving up property claims, or demanding to secure them, are a mark of the gender-based power of the parties involved and the circumstances of the divorce. In general, divorce implicates a switch from amity to enmity fueled by senses of protest and imputations of fault. Where children are involved, complex and potentially bitter conflicts can ensue about legal claims and legal responsibilities, and all these litigious procedures can also be seen as ritual processes translating ambiguous complexities of relationship into digital answers. As with the Maring people of Papua New Guinea's feasting practices of the kaiko analyzed by Roy Rappaport (Rappaport 1968), the function of these legal rituals is to shift a social situation that has entered conflictual ambiguity (a wavering analog state) and shift it back into new certainty (digital state) that creates a stable structure out of uncertainty. We mean this in a purely legal sense since interpersonal turbulence can still occur even after a legal settlement.

Nevertheless, stability is the ritual aim of such court-based and law-based decision-making. This, in turn, can be upended if separate jurisdictions are involved and legal arrangements become subject to dispute.

Custody rules regarding children form a particularly contentious arena following a divorce if the children have not attained adulthood, and questions of financial support are often crucial, again up to a certain age. State-based legal procedures deal with such questions in a way quite different from customary consensual arrangements, such as we find in the Papua New Guinea Highlands. A wife in parts of Papua New Guinea may suggest that she and the spouse should themselves divide up the children and take care of them separately with or without aid of their respective kin. If such an informal method of decision-making fails, the parties may go to whatever local bodies can attempt to adjudicate their competing claims. Such bodies include village courts that operate on customary levels or district courts presided over by salaried magistrates appointed by central government bodies in legal service or ministries. Such magistrates can also adjudicate regarding child support. It goes without saying that issues of this kind arise because of intrusion of monetary needs into society, which is not found in subsistence-based systems. In the latter, kinship ties take care of most needs and people have control of their own modes of production and consumption. The most immediate ways in which subsistence systems become unable to cope in customary terms are the result of state imposition of taxes and school fee payments.

Churches in Papua New Guinea attempt to play a significant role in family welfare and divorce issues. Their workers may advise against divorce and advocate long-term monogamy along with a Christian mode of life. For Catholics, divorce as such is generally not permitted, but annulments declaring that a marriage is void can be set in hand. Entry into marriage for Seventh Day Adventists is easier in bridewealth societies because the Adventists do not require adherence to such marriage payments. They do, however, advocate monogamy and adherence to church rules, such as Saturday worship. All in

all, Christian churches discourage divorce, and it is notable that, in the English Anglican Church, divorced people who seek to remarry in church cannot do so and the 'blessing' they can receive in church is technically an admission of 'repentance' that their former marriage had broken down. Churches have contradictory rules. The Presbyterian-based Church of Scotland does allow divorced persons to marry in church, for example, while Catholics do not allow divorce as a general rule. Similar multiplicity exists in relation to same-sex marriage, with, no doubt, development of case law to cope with particularities. It is of interest to note that ownership or custodianship of pets can enter the domain of disputes in relation to whether they are to be considered as parts of a conjugal fund or should be seen as the property of one or the other spouse. Pets are the nearest thing to the humans who keep them and may be regarded as 'children' in the context of divorce.

### 3. Social Structures

One issue in divorce studies is how frequency of divorce is related to broader questions of social structure. A standard approach to this question within the domain of kinship is that there is a difference between societies with matrilineal descent groups and those with patrilineal group forms. The difference is played out specifically in terms of gender. Matriliny provides children an affiliation with their mother's group, so the mother is not bound to the husband in the same way as in patrilineal structures going with the institutions of bridewealth payments. In matriliny, paternity may be important, but it can also be relatively minimal. Consequently, in divorce, where children are affiliated correlates broadly with the mode of descent. There are variations, of course. The flexibility with which possibilities of divorce are regarded can vary in accordance with sexual mores.

The ideology of the Nuer people in South Sudan that cattle beget children means that children can remain affiliated in the father's group even if the wife has gone elsewhere and is living with another partner (Evans-Pritchard 1951, p. 122). Alternatively, such a partner may make a special payment in cattle to claim a child as his own. The extreme historical case of warrior-caste groups among the Nayar of Kerala in southern India reveals that women there could take multiple sexual partners while their children invariantly belonged to the matriline regardless of paternal connections. This rule led to the assessment that marriage itself was 'minimal' among the Nayar. This sketch is sufficient to show that social structure does greatly influence divorce (see Gough 1961; Strathern and Stewart [2011] 2016).

If we turn back to how contemporary circumstances bear upon incidence of divorce in Euro-American society, one point is that pre-nuptial agreements, or pre-divorce agreements as they could be called, can problematically cross-cut operation of conjugal fund rules and, consequently, themselves set up negative feelings between couples. Another is that contraception means that sexual activity is not necessarily bound with procreation of children. Moreover, people move around and meet diverse others, leading to spousal separation and divorces. Inequities in wealth weaken valency of marital ties. Rules of no-fault divorce make divorce easier. The influence of churches is diminished by processes of secularization in society. Patriarchy, thus, does not rule. Altogether, social structure has become 'looser' than it was at times in the past, analogous to debates about transformations in kin structures in southeast Asia and Oceania stemming from studies in the 1960s. Loose structure means more variation, either in affiliation or in marriage. In this way, divorce must be seen as a symptom of wider forces of change outside the realm of kinship studies but impinging all the same on kinship. Ethnographies of marriage are, therefore, bound up with issues of divorce.

We add here a more detailed account of divorce among the Nuer.

### 4. Divorce among the Nuer of South Sudan

Theorizing about divorce in relation to gender and identity in societies with nominally patrilineal descent groups preoccupied the attention of anthropologists working in the traditions of descent and alliance theory in the mid-twentieth century onward. Detailed

holistic ethnographic studies can show the complexities of practices in societies with unilineal descent, indicating how 'rules' can be subverted or reinterpreted, including operation of marriage ties and also their dissolution, along with the practices of concubinage outside the forms of marriage as such. Anthropologist Evans-Pritchard, who lived among the Nuer for varying periods of time from 1930–1936, records the difficulties of carrying out his fieldwork amongst them. He found that, in political terms, structured unilineal groups were important, while, in daily domestic life within settlements and cattle camps seasonally occupied, horizontal links traced through women were also important, tying together disparate lineages by means of marriages and the relationships of filiation that such marriages created. In a local community such as a cattle camp, everyone in it could trace some sort of kinship tie (mar in the Nuer language), while, at the same time, the community was divided into separate clans and lineages defined in patrilineal terms or else incorporating through female ties numbers of immigrants and adopted persons, some even from a different ethnic group, the Dinka incorporated into Nuer groups as a result of warfare. The particular feature that enabled accretions of various kinds to the structure of local comminates was that each group had as its core members of agnatically related persons, around whom clusters of agnates could gather. The core members could be described as 'bulls' of the lineages and, among them, some stood out individually as leaders.

Cattle were an important means whereby reproductive relationships were structured, whether by a recognized form of marriage or widow-concubinage or concubinage outside of marriage. Evans-Pritchard (1951, p. 26) further notes that unmarried women may have children by particular partners, and sons that are born of these ties can become legal children of these male partners through payments of cattle in the same way as for formal marriages. Aside from this point, men sometimes choose to live with their wife's people, and, as a result, their children come to belong to the wife's community and they too are content to stay in the wife's place, although they can always decide at a later time to return to their agnatic place.

The crucial point to recognize here is that marriage among the Nuer was a jural or 'legal' contract founded on payments of cattle over time to the bride's people, which were designed to secure the affiliation of children borne by the bride to the lineage or individual who provided the cattle and paid them. It is necessary to understand that this one fact makes all the difference to the issue of divorce among the Nuer. A wife could leave her husband and go live with another man, even taking her children by the husband with her. She might subsequently bear children by her new partner. Eventually, however, all those children would be deemed to belong to the lineage of the husband who had originally paid the cattle of bridewealth. The presumed genitor of the children may also make claims on them by payments of cattle, beginning with a fine for adultery. Such a fine would be returned in part later if a healthy child were born of the adultery. Out of a nominal number of six cows initially paid, five would be returned. This arrangement Evans-Pritchard (1951, pp. 121–22) explains as follows, based on the practices of the western Nuer: "It is believed that if the cattle are not returned, the child may die, and his ghost haunt his pater" [the husband] (p. 121). "The sixth cow, the one that remains with the husband, is called yang kula, the cow of the hide. The payment protects the husband from the sickness he might otherwise suffer were he to have relations with his wife when the adulterer has had relations with her in the same sleeping-hide, as the Nuer put it" (p. 121). That is, the sixth cow paid by the adulterer to the husband may be seen as placating the lineage ancestors of the husband since, otherwise, they would be affronted by these 'out of place' sexual relations, with the wife endangering the husband in his own settlement place or home, if we may interpret the adultery as the breaking of a spatial taboo (see Douglas 1966). From the Nuer viewpoint, adultery committed in the husband's place is serious because of the potent character of bodily sexual emissions. Of course, none of this applies if the adultery takes place 'in the bush', away from habitation. Still less is it of concern if the wife actually goes to live with another man elsewhere. However, if she bears children in this new place,

her legal husband can claim these children as an entailment of the bridewealth rule that 'cattle beget' children' Thus, divorce among the Nuer takes on a very different aspect from divorce in a simple patrilineal model based on requirements of fidelity by the wife, or in a culture in which sexual relations outside of marriage initially become a powerful cause of divorce.

Evans-Pritchard (1951, pp. 91–96) makes clear in general the circumstances under which divorce could occur among the Nuer. Bridewealth in payments of cattle is very important in benefiting the bride's kin, who, therefore, have a strong interest in preserving the marriage. However, if no children are born of the marriage, the marriage itself can be dissolved by return of all the bridewealth cattle except for two beasts kept by the bride's parents in recognition that the marriage was consummated even though no children were born as a result (Evans-Pritchard 1951, p. 51). The situation is different if children have been born. In that case, the husband and his kin may decide to forego the return of the bridewealth cattle and "may leave all the cattle she may 'bear in the bush', by lovers. She is then not divorced, but she is only separated from her husband, and she cannot remarry" (ibid.). "On the other hand, if the husband's kin claim all the bridewealth back, apart from six head of cattle which the husband leaves with the wife's people to hold his claims onto the sons that may have been born, then the wife by this procedure is divorced and can remarry" (ibid.). If the wife has borne two children and nevertheless leaves her husband, the bridewealth cattle are not repaid, the children are claimed by his kin, and any children she later bears by other men are also claimed by the husband, and she is not divorced.

Much depends on whether the couple have been on good terms with each other and also on the husband's relations with the wife's kin. If these relations are not good and she has run away to her parents and they are unwilling to return the bridewealth cattle, they will tell the husband to come and take her back, but, most likely, this will not happen, and he can only claim any further children she may subsequently have.

Two factors are involved: bridewealth and affiliation of children. Payment of bridewealth does not by itself stabilize a marriage since, until a child is born, the marriage process remains incomplete (Evans-Pritchard 1951, p. 93). Until there is a child, "the wife is considered to belong to her own kin and not to her husband's kin" (loc. cit., p. 94). In any case, the bridewealth payments are not expected to be complete at this time. When they are complete and a child is born, once the child has been brought to the husband's home from the wife's parents' place and she has stayed with the husband "for a year or two" (p. 94), Evans-Pritchard observes that divorce would be very unusual. He also observes that the relatives on both sides would do their best to maintain good relations. This process extends over time as children are born and life-cycle payments continue (exactly as in the Papua New Guinea Highlands).

Evans-Pritchard's discussion makes it clear that transfers of bridewealth cattle determine what we can call structured or jural aspects of marriage and sexual relationships. In experiential terms, however, it is interpersonal goodwill that influences action. The example of the Nuer is a telling counterpoint to ideas about divorce in contemporary industrial societies. Bridewealth makes a large difference in jural rules, but it does not determine experiential processes, as witnessed by the practices of concubinages. In cultures where sexual infidelity is expected to be portrayed as a cause of marriage breakdown and divorce, its outcomes are very different from how it is, or was, regarded by the Nuer, with whom bridewealth determines the existence of the marriage in structural terms, but women can practice experiential ties as they see fit, at the same time not being able to become divorced and remarry except when all the bridewealth paid for her (or almost all) has been repaid. She may choose concubinage at will but not whether she is still married to a husband and his lineage. It is the whole ritual complex surrounding bridewealth that determines this. Thus, we see that ritual defines structure but not personal experience.

The case of the Nuer became famous in anthropology because of the remarkable twist their practices place on operation of unilinear descent in their social structure, in which such descent governs politico-jural relations among groups, but bridewealth has further

specific consequences at the domestic or familial levels. If this fact is considered, similarities become clear between the Nuer and, for example, the Mount Hagen people in the Papua New Guinea Highlands. John Barnes (Barnes 1962) early on argued against application of African models to New Guinea Highlands societies, but, in fact, illuminating comparisons can be made when we look at more detailed accounts.

Our own discussion here reveals, per contra, that the Nuer model, at any rate, helps us to understand New Guinea realities. In the sphere of divorce and its rituals, we can also discern similarities and differences, specifically deriving from the way bridewealth practices and conjugal fund norms operate in the jural sphere of reproductive processes. In general, because marriage is marked by rituals, divorce must also be marked with its rituals of reversal of marriage rites. The constant feature is the importance of ritual in creating and altering social relations and, at the same time, the ingenuity with which human propensities are flexibly accommodated within broad structural parameters.

**Author Contributions:** The two authors jointly conceptualized, wrote, and edited this article. All authors have read and agreed to the published version of the manuscript.

**Funding:** This research received no external funding.

**Conflicts of Interest:** The authors declare no conflict of interest.

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
