# Peer review of "Divorce: Experiential and Structural Elements: Cases from Papua New Guinea and Africa"

_religions, doi:10.3390/rel14030303_

Round 1

Reviewer 1 Report

The title should indicate that the study is about New Guinea and Sudan.

Author Response

Thank you and good wishes.

Reviewer 2 Report

Please see below

Author Response

Thank you and good wishes.

Reviewer 3 Report

This article presents a very intriguing aspect of how divorce is perceived among the Nuer. It makes a signficant contribution to academic research, especially on the interface of religion and culture. However, there are minor content issues needing attention. I have highlighted these via track changes.

Author Response

Thank you very much. All changes addressed.